# Challenging Cellular Homeostasis: Spatial and Temporal Regulation of miRNAs

**DOI:** 10.3390/ijms232416152

**Published:** 2022-12-18

**Authors:** Naomi van Wijk, Keren Zohar, Michal Linial

**Affiliations:** Department of Biological Chemistry, The Alexander Silberman Institute of Life Sciences, Faculty of Sciences, The Hebrew University of Jerusalem, Jerusalem 91904, Israel

**Keywords:** pre-miRNA processing, RNA degradation, translation control, stress granules, P-bodies, oxidative stress, Ago, RISC, viral infection

## Abstract

Mature microRNAs (miRNAs) are single-stranded non-coding RNA (ncRNA) molecules that act in post-transcriptional regulation in animals and plants. A mature miRNA is the end product of consecutive, highly regulated processing steps of the primary miRNA transcript. Following base-paring of the mature miRNA with its mRNA target, translation is inhibited, and the targeted mRNA is degraded. There are hundreds of miRNAs in each cell that work together to regulate cellular key processes, including development, differentiation, cell cycle, apoptosis, inflammation, viral infection, and more. In this review, we present an overlooked layer of cellular regulation that addresses cell dynamics affecting miRNA accessibility. We discuss the regulation of miRNA local storage and translocation among cell compartments. The local amounts of the miRNAs and their targets dictate their actual availability, which determines the ability to fine-tune cell responses to abrupt or chronic changes. We emphasize that changes in miRNA storage and compactization occur under induced stress and changing conditions. Furthermore, we demonstrate shared principles on cell physiology, governed by miRNA under oxidative stress, tumorigenesis, viral infection, or synaptic plasticity. The evidence presented in this review article highlights the importance of spatial and temporal miRNA regulation for cell physiology. We argue that limiting the research to mature miRNAs within the cytosol undermines our understanding of the efficacy of miRNAs to regulate cell fate under stress conditions.

## 1. miRNA Function at the Right Place and the Right Time

Since the discovery of the first microRNA (miRNA) some three decades ago, thousands of additional miRNAs regulating gene expression in multi-cellular organisms have been identified. Many of the principles and rules for the pairing of miRNAs with their targets have been validated experimentally [1]. Based on experimental CLIP-seq results and detailed structural information, some of the rules for miRNA-target interactions were revised, including non-canonical pairing, binding beyond the 3′-UTR, and more [2,3]. Unfortunately, currently available miRNA-target identification tools and algorithms fail to address the complexity of the miRNA–mRNA regulatory network [4]. Within living cells, intense competition exists for free and bound miRNAs [5]. Moreover, bounded miRNAs are not limited to their direct interaction with mRNA targets. miRNAs can be bound by other RNAs, such as long non-coding RNAs (lncRNAs), pseudogenes, or circular RNAs (circRNAs), and by RNA binding proteins. The binding of miRNAs to any of these competitors by ‘sponging’ dominates miRNA accessibility and potency [6,7]. It is anticipated that the quantitative measurements of binding affinity, amounts, and stoichiometry between miRNAs and mRNAs within cells are critical for the assessment of the cellular regulatory network and its dynamics [8]. Therefore, a major drawback in miRNA research concerns the difficulty to link molecular knowledge to our understanding of miRNA function at the cellular and intercellular levels. Herein, we aim to provide an integrative view of spatial and temporal aspects of miRNA function under homeostatic and stressed conditions.

We present an overview of miRNA quantities in humans followed by a discussion of the spatial and temporal organization of miRNAs. We then consider the compartmentalization of miRNAs and their targets in undisturbed cells and in cells exposed to stress. More specifically, we summarize the evidence for miRNA function beyond the cytosol by considering the availability of miRNAs in subcellular localizations including the nucleus, mitochondria, and the ER, as well as in membrane-less compartments. In addition, we review evidence on temporal changes in the localization of miRNAs and their targets in response to environmental stress. We highlight settings in which translational control is critical to resolving special regulatory needs as in the case of viral infection, as well as the role of miRNAs in intercellular regulation as delivered by exosomal communication.

## 2. Biogenesis and Diversity of miRNAs: Rules and Exceptions

Since the discovery of the first miRNA molecules and the RNA silencing machinery, the identity and amounts of miRNA in mammals and other organisms remain one of the main questions in this regard. How are miRNA genes distributed along the genome and how are they transcribed? How do miRNAs maturate and what post-transcriptional processes create the diversity of miRNA molecules? Mature miRNAs are single-stranded RNA molecules of ~22 nucleotides (nt), which comprise the largest collection of non-coding RNA molecules (ncRNAs) in human cells. In the human genome, there are >1900 miRNA genes that account for approximately 2600 mature miRNAs, many of which are still waiting for validation [1,9,10]. Many miRNAs are located within the intronic sequences of protein-coding host genes. Notably, the expression of the host gene and the miRNA are not necessarily coupled, and miRNAs are transcribed by their own promoters. While most miRNAs are transcribed by RNA Polymerase II, miRNA genes located adjacent to Alu repeats can be transcribed by RNA Polymerase III [11].

The biogenesis of mature miRNAs includes a series of enzymatic cleavages, modifications, and strand selection [12,13]. In short, the primary transcript (pri-miRNA, ~200 nt) is cleaved and processed by the activity of a microprocessor complex in the nucleus, which includes the gene products of *DROSHA* and its cofactor *DGCR8*, to create a stem-loop structure called pre-miRNA (~70 nt). The pre-miRNA is then actively transported to the cytoplasm by the conserved nuclear exporter Exportin-5 [14], where it is further processed to a functional mature miRNA by the catalytic activity of Dicer [12,15]. Usually, only one of the Dicer-cleaved strands becomes functional and is further stabilized by tightly binding to Argonaut (Ago) proteins within the micro-RNA-induced silencing complex (miRISC) [16,17,18]. The 5′ sequence of mature miRNAs includes a seed of 6–8 nucleotides that anchors the miRNA to the miRNA binding site (MBSs), canonically located at the 3′-UTR of target mRNA [5,19,20]. Of note, many miRNA binding sites were found to be located in the target’s 5′-UTR or protein-coding sequences (CDSs) [21,22]. The bound transcript inhibits protein translation by interfering with initiation, elongation, or termination steps [18,23]. The targeted transcript itself may undergo degradation and further processing, such as deadenylation or decapping [19,24].

The number of miRNAs within living cells is highly regulated at transcriptional and post-transcriptional levels. As a result of non-canonical biogenesis, the number of different miRNAs is further increased [25]. The large collection of miRNAs in humans is not only a reflection of the large number of genes but also a result of the canonical and non-canonical formation and stabilization of functional miRNAs. The collection of known miRNAs has been expanding rapidly in recent years, reflecting the increased sensitivity of deep sequencing and the development of sensitive CLIP-seq technologies [26,27]. While many miRNAs are conserved and share features across the animal kingdom [28,29], we adhere to the current knowledge from human cells and tissues. Figure 1 presents a global view on the features of human miRNAs as derived from the most comprehensive collection of miRBase databases to date [30]. It is evident that only a quarter of the miRNAs are linked to experimental evidence (Figure 1B). Different miRNAs can ranges of 4–6 orders of magnitude in a single cell [30], with two-thirds of the observed miRNAs expressed at low levels (Figure 1C). The spacing of miRNAs is skewed, with about 340 miRNAs in close vicinity to other miRNAs (<10 k), where the rest of the miRNAs are quite far from other miRNAs (Figure 1D). In addition, miRNAs vary drastically in their affinities and specificities.

According to the simple view of miRNA biogenesis, there is only one guided strand that will act as a mature miRNA in the cell. In reality, the number and identity of the miRNAs within living cells are highly regulated. Numerous DNA binding proteins including tumor suppressors (e.g., Breast cancer type 1 susceptibility protein; BRCA1), transcription factors (e.g., p53), and cell helicases (e.g., DHX9) can affect the accuracy and efficacy of the microprocessor [31]. Moreover, the interactions of Dicer with conserved RNA-binding proteins (e.g., RISC-loading complex subunit TARBP2) dictate the fidelity of the strand selection process [32]. Figure 2 illustrates a sample of expressed human miRNAs, emphasizing the number of the mature miRNAs (i.e., derived from the pre-miRNA 3′, 5′, or both). In addition, the editing and modification of the cleaved product increase the diversity of miRNA sequences (isomiRs) [33]. In general, the final product of the miRNA processing, the extent of isomiRs, and their stability are governed by the cell-dependent accessory proteins that interact with the main processing machinery (i.e., microprocessor, dicer, and Ago).

In summary, the number and diversity of miRNAs are highly regulated both through transcription, post-transcriptional modifications including processing to mature miRNA, and turnover, enabling their function in the regulation of mRNA translation.

## 3. How Are miRNA Dynamics Studied?

In addition to the determination of miRNAs quantities in the cell, it is of importance to determine where they are located. Which methods enable the study of spatiotemporal aspects of miRNA dynamics? Can we determine these dynamics at the single-cell or even at the single-molecule level? In this section, we briefly discuss some of the experimental approaches that drive the current knowledge of the field.

### 3.1. Fluorescent In Situ Hybridization of miRNAs

Various fluorescence-based methods have been used to visualize miRNAs. One of the most used approaches is RNA fluorescent in situ hybridization (FISH), where fluorescently tagged antisense molecules are used for the detection of specific RNA sequences. Surface and intracellular protein markers can be stained with fluorescently labeled antibodies for simultaneous protein and mRNA imaging. Using flow cytometry, cells can be sorted based on antibodies that recognize markers of a specific subpopulation of cells. This method enables the study of the localization of mRNA and miRNA at the single-cell level.

In FISH, the mRNA or miRNA of interest is hybridized with an antisense sequence tagged with a fluorescent tag [34]. A highly sensitive FISH method based on branched DNA uses tagged antisense sequences with different excitation/emission spectra to simultaneously detect miRNAs and mRNAs and their cellular localization through imaging. When applied for miR-21 and miR-155, this method revealed the upregulation of miR-155 upon T cell activation [35]. Similarly, the miRNA probe can contain locked nucleic acids (LNAs), which are bicyclic high-affinity RNA analogs that increase hybridization specificity. This is especially relevant for the in situ hybridization of miRNA because a higher annealing temperature (T_m_) between the complementary RNA strands enables the use of very short (~20 nt) probes while retaining high specificity [36]. The main disadvantage of FISH is the potential for impeding the function of the miRNA upon hybridization with the antisense fluorescent probe. Direct miRNA fluorescent tagging may overcome this drawback, as was demonstrated in a method called intracellular single-molecule, high-resolution localization and counting (iSHiRLoC). In iSHiRLoC, fluorophore-labeled miRNAs (or other small RNAs) are microinjected into the cell and single particle tracking (SPT) microscopy enabled the study of their dynamics, localization, strand selection, and turnover of specific miRNA molecules. The application of this method was demonstrated for let-7a-1 and for artificial miRNA tagged with a Cy3 or Cy5 fluorophore [37].

In summary, a combination of advanced, high-resolution imaging approaches has greatly improved miRNA detection and miRNA dynamics within cells.

### 3.2. Cellular Fractionation

In addition to the whole-cell approach described above, how can we specifically study the miRNA composition in their subcellular organelles and structures? An approach that has been in use for many decades is cell fractionation, in which organelles of interest are isolated and the miRNAs present in that fraction are subsequently assayed. Cell fractionation is based on the relative biophysical properties of the organelle of interest such as size, density, and unique markers of different cell compartments. Classical fractionation that partition organelles rely on differential centrifugation, density gradients, size excluding columns, and affinity enrichment. The goal of all the fractionation protocols is to generate high-resolution fractions at increasing purity [38,39,40]. For example, Li et al. used fractionation to separate nucleoli from the nucleus to study the dynamics of miRNAs such as miR-191, miR-193b, miR-484, and miR-574-3p in HeLa cells. They showed that the nucleolar association of these miRNAs is resistant to cellular stresses but sensitive to exogenous nucleic acids. Introducing double-stranded DNA or RNA causes miRNAs’ redistribution to the cytosol [41]. In another example, mitochondria were purified using cell fractionation and subsequent immune isolation. A comparison between miRNAs found in mitochondria and in the cytoplasm yielded over 50 differentially localized miRNAs that show significant enrichment in mitochondria [42]. Following fractionation, the isolation and identification of miRNAs and their abundance in each fraction can be performed by Northern blot, RT-qPCR, direct sequencing, or DNA microarray.

### 3.3. Single-Cell Dynamics

The use of single-molecule FRET (smFRET) technology allows the monitoring of conformational states and dynamics of Ago2-RNA complexes in solution. Specifically, testing the conformational flexibility of Ago2 and pre-miR-451 provided detailed information on the dynamics and lifetime of the ribonucleoprotein particles. The sequential events by which miRNA-guide RNA directs Ago to base-pair with the target RNAs and causes translation inhibition followed by mRNA decay have been demonstrated [43]. Several creative single-molecule tracking techniques have been developed for miRNAs. For example, a dual-fluorescence reporter system was introduced into mouse embryos in utero for monitoring the dynamics of specific miRNAs in live cells [44]. Additionally, time-dependent fluorescence spectra with FRET-based localized probes use hairpin-based DNA nanostructures for rapid, efficient, and reliable imaging of intracellular miRNAs [45]. Such single-cell technologies allow for the monitoring of miRNA molecules with exceptional spatial and temporal resolution.

## 4. Cellular Homeostasis and miRNA Availability

Research on miRNAs is often discussed at different resolutions, from organism and tissues to individual cells. In organismal development, the role of ncRNAs in general, and miRNAs particularly, has been extensively researched. MiRNAs have been shown to play a role in almost all aspects of animal embryogenesis [46,47,48]. For example, in zebrafish and frogs, the mature miR-430 accelerates the decay of hundreds of maternal RNAs, allowing maternal–zygotic transition. The human orthologues of miR-430 (miR-302, miR-372, and miR-516–520) play a role in the maintenance of embryonic stem cell pluripotency [49]. From invertebrates to humans, ncRNAs play an important role in the patterning of body axes and determining the fate of cells and tissues. For example, miR-124 is a master regulator of neural development. Its role in mammals is manifested at various levels of neuronal differentiation, including splicing, chromatin remodeling, and gene expression. In adult mice, miR-124 has also been shown to promote neurogenesis [47].

One of the main questions is how miRNA localization controls cellular homeostasis. Which processes are regulated by miRNA? How are miRNA regulatory networks of these cellular processes affected by interactions with other RNAs that may reduce or increase their local concentration? Under steady-state conditions, miRNAs act as molecular agents for attenuating transcriptional noise [50]. Specific miRNAs are expressed in limited tissues and cell types, for example, miR-122a in liver tissue, miR-1 and miR-133a in heart and skeletal muscle cells, and miR-9 in the brain [51]. Furthermore, a combined experimental and computational approach demonstrated that some abundant miRNAs affect the cell state much more than others. Furthermore, the translational machinery is more robust and quite immune to changes in miRNA levels as opposed to mechanisms such as transcription that are sensitive to the precise cellular miRNAs’ profile [52].

The miRNA profile is established to maintain cell state [8], metabolic homeostasis [53,54], proliferative potential [55], and stemness [56]. For example, miR-628-5p negatively regulates mesenchymal stem cell (MSC) stemness during periodontal regeneration. Periodontal regenerative cytokines act as miR-628-5p suppressors to support periodontal regeneration [57]. Metabolic homeostasis is also highly dependent on miRNA regulation. For example, the homeostasis of HDL and cholesterol is tightly regulated by the highly conserved miR-33 family, encoded within an intron of *SREBP-1* and *SREBP-2*, and acting upon *ABCA1* mRNA, thus affecting intracellular cholesterol efflux between the liver and peripheral tissues [58,59]. Finally, proliferation is tightly controlled by miRNAs that act upon the major regulators of cell cycle progression such as let-7, which, in combination with other miRNAs, regulates Cyclin D, CDC25a, and CDK2/4/6, and regulates entry into the G1 phase, whereas it also regulates CDC34, CDC25a, and Cyclin A, regulating entry into the G2 phase [60].

RNA localization is known to drive development and differentiation, but has received little attention in the context of miRNA regulation [61]. The full potential of miRNAs in cells must be discussed from both a probabilistic and quantitative standpoint [52,62]. Achieving cell homeostasis is a result of the non-linear interaction of multiple miRNAs according to their availability [8,63]. The contribution of spatial and temporal considerations for deciphering miRNA regulatory networks [64] was studied by the crosstalk of miRNAs with long non-coding RNAs (lncRNAs), small nuclear RNAs (snRNAs), and circular RNAs (circRNAs). Many lncRNAs and circRNAs were shown to contain multiple miRNA-binding sites, thus acting as ‘sponges’ of miRNA, competing with mRNA binding. This concept of competing endogenous RNA (ceRNA) was proposed a decade ago and has had a considerable impact on our understanding of the miRNA regulatory network [65]. This mechanism reduces local miRNA concentration and ability to act upon their target mRNAs. For example, miR-204 is sponged by lncRNA MALAT1, causing upregulation of Smad4, thus positively regulating osteoblast differentiation in calcific aortic valve disease [66]. In another example, pri-miRNA processing is localized to nuclear paraspeckles where it is bound to NEAT1, a lncRNA that interacts also with many RNA binding proteins, including the NONO/p54 heterodimer [67]. Many circRNAs also function as miRNA sponges [68]. For example, circ-NRIP1 sponges miR-149-5p, causing the upregulation of AKT1 and consequently the stimulation of tumor progression in many cancer types [68]. Additionally, miR-145 regulates vascular smooth muscle cell (VSMC) physiology and homeostasis and its modulation is associated with atherosclerosis and arterial restenosis. It is bound by circ-lrp6, which is enriched in VSMCs, mostly in cytoplasmic membrane-less loci called processing bodies. The circ-lrp6 regulates miR-145 availability by sponging mechanisms and, thus, controls migration, differentiation, and proliferation in VSMCs [69].

It is thus clear that for the regulation of homeostatic cellular processes, not only are the identity and sequence of the miRNAs important, but also their availability to act upon their targets. This regulatory layer of RNA–RNA interactions requires further investigation and should be considered when studying miRNA-related pathologies.

## 5. Beyond the Cytosol: miRNA Regulates Organellar Function Locally

A large body of research has studied the function of miRNAs in repressing cytoplasmic targets. However, research from the last two decades has increasingly shown that the highly regulated subcellular compartmentalization of miRNAs is essential for their function [70]. Are miRNAs located in all organelles beyond the cytoplasm? Are miRNAs located in the organelles whose function they regulate? What non-canonical functions do miRNAs fulfill in the organelles to which they are localized? In this section, we briefly review the localization of miRNAs in the nucleus, mitochondria, and ER, as well as in the membrane-less compartments [71,72].

### 5.1. The Nucleus

A large number of miRNAs were isolated from biochemically purified nuclei. Among them, some are exclusively found there or substantially enriched in cell nuclei [73]. Accumulated evidence for nuclear localization argues for potential roles in transcriptional and post-transcriptional regulation as studied in the context of cancer, immunity, and chronic disease [74,75]. For example, miRNAs can act directly on cis-regulatory elements (CREs), either suppressing or activating gene transcription. Some miRNAs are co-localized to sub-nuclear structures such as the nucleolus, open chromatin [76], and the spliceosome [77]. Mature miRNAs and other pre-RNA-derived sequences can compete for splicing events co-transcriptionally [78]. In addition, miRNAs were proposed to affect epigenetic changes [79], impacting chromatin structures in the nucleus and consequently the cell’s transcriptional potential. An open question concerns the shuttling of mature miRNAs from the cytosol back into the nucleus. It is not known whether this dynamic is driven by nuclear localization signals or by the positioning of mRNA targets.

Once miRNAs were discovered to be active components of cell regulation, including epigenetics and transcriptional regulation, evidence was found for additional surprising functions. For example, miRNA in the nucleus acts as an antisense by blocking the RNA polymerase binding site [80], masking promoters, functional regions, and splicing sites [77]. This was for example demonstrated for miR-665, which is enriched in the nucleus of cardiomyocytes in humans and mice. In the nucleus, it binds to a conserved binding site in the *PTEN* (phosphatase and tensin homolog) promoter and suppresses its transcription. As a result of the direct inhibition of *PTEN* expression, heart failure is exacerbated [81]. The hepatic nuclear miRNA-122 directly controls the level of miR-21 by binding to its transcript and preventing the DROSHA-DGCR8 microprocessor’s conversion of pri-miR-21 into pre-miR-21 [82].

Furthermore, a direct regulatory link exists in the nucleus between miR-9 and the abundant lncRNA MALAT (metastasis-associated lung adenocarcinoma transcript 1). In the nucleus, the binding of miRNA to complementary sequences present in MALAT1 is Ago2-dependent [83]. This example confirms the presence of post-transcriptional regulators in the nuclear compartment and extends the direct regulation of miRNAs on other ncRNAs.

### 5.2. The Mitochondria

Mitochondrial DNA (mtDNA) is 16 kbp long and encodes only a dozen proteins. However, a small number of miRNAs are encoded in mtDNA, and in addition, several nuclear-encoded miRNAs are translocated into the mitochondria (mito-miRNAs/mitomiRs) [84]. A total of 150 miRNAs have been detected in mitochondria, of which the most frequently identified miRNAs include let-7b-5p, let-7g-5p, miR-107, miR-125b-5p, miR-181a-5p, miR-221-5p, and miR-494-3 [85,86]. These mitochondria-localized miRNAs are associated with Ago2 and are likely to be involved in the modulation of mitochondria-specific functions [87,88], including the electron transport chain (ETC), tricarboxylic acid (TCA) cycle, β-oxidation, and lipid and amino acid metabolism [42,89]. For example, as many as ten miRNAs regulate the expression of citrate synthase, which catalyzes the first step in the TCA cycle. ATP synthase, the mitochondrial complex generating the cell’s ATP, is targeted by miR-338-5p and a set of mitomiRs (i.e., miR-127-5p, miR-101-3p, miR-181c-5p, and miR-210-5p).

Due to the importance of mitomiRs in mitochondrial translation and transcription, a direct link was made with a number of diseases involving cellular metabolism or the cell’s ATP balance, such as cancer, diabetes, and obesity [90,91]. As expected, the role of mitochondrial regulation in cancer progression is often mediated by mitomiRs, the levels of which dictate metabolism and apoptosis in various cancer types [92,93]. For example, miR-4485 regulates the processing of pre-rRNA of the mitochondrial 16S rRNA, and affects ATP production, reactive oxygen species (ROS) levels, apoptosis, and the function of mitochondrial complex I in breast carcinoma [94]. An emerging role of miRNAs in the initiation and progression of chronic diseases, including cardiovascular diseases and metabolic diseases, is attributed to mitomiRs. For example, the upregulation of miR-181c, which targets Cytochrome c oxidase I (mt-COX1) mRNA, is related to heart failure, type 2 diabetes, and aging [95]. In contrast, the downregulation of miR-696, which targets the mRNA of peroxisome proliferator-activated receptor gamma coactivator 1-alpha (PGC-1α), a key regulator of energy metabolism, is implicated in metabolic disorders [96]. The RISC formed by miRNAs involved in mitochondrial function is formed mostly in the cytoplasm, but some studies have demonstrated RISC formation in the mitochondria [97].

Some miRNA-encoding genes contain open reading frames that are translated to short proteins (miRNA-encoded peptides, miPEPs) [98]. For example, the gene encoding miR-34 is translated into a microprotein of 133 amino acids, called miPEP-133. In healthy tissue, this peptide has a tumor suppressor function, whereas it is downregulated in various cancer cell lines. It directly binds the mitochondrial heat shock protein 70 (HSPA9) in mitochondria [99]. Interestingly, miPEP-133 enhances the transcription of its own pri-miRNA. Although several additional miPEPs have been found, including one encoded in pri-miR-200a/b, the generality of such a positive autoregulatory loop is still questionable in humans [100].

### 5.3. The Endoplasmic Reticulum (ER)

The endoplasmic reticulum (ER) is emerging as a major location of miRNA-mediated regulation [101]. The ER is a hub for translational control of the secretory proteome, the translational efficiency of which is tightly controlled and coordinated in time and space [102]. Moreover, the surface area in the ER is strongly correlated with translation efficiency and protein trafficking (e.g., in endocrine and exocrine cells). The result is that specific mRNAs located in the ER membrane are exposed to a distinct collection of regulatory factors and translational components. In such cases, mRNAs can be locally suppressed in the vicinity of the ER [101]. Moreover, the miRNA biogenesis machinery has been found to localize at close proximity to the ER. For instance, the recruitment of Dicer to the ER occurs most likely through CLIMP-63, a protein of the cytoskeleton-linked ER membrane [103].

From a cellular perspective, translation occurs with free or ER-bound polysomes, and miRNA compartmentalization can shift the balance. For example, the trafficking of Ago2 acts as a rate-limiting step in miRISC formation on ER-attached polysomes. Specifically, the ER-mitochondrial tethering controls Ago2 availability. Interestingly, the membrane potential of the mitochondria acts as a sensor for Ago2 trafficking and consequently for miRNA stability [104]. Specifically, the mRNA localized to ER-bound polysomes is a prerequisite of Ago2-miRNA binding. The suppression of translation by miRNA at the rough ER membrane is sequential and occurs after the initiation of translation [105].

In summary, miRNAs located in close proximity to the nucleus, mitochondrion, and ER fulfill specific functions that are absent in the cytoplasm. The nucleus is not only the site of miRNA transcription, but mature miRNAs are transported back and directly act upon nuclear targets to execute nuclear functions including splicing, transcription, and regulation through the ‘sponging’ of other RNAs. In mitochondria, although many miRNAs act upon organelle-specific functions, only a few originate from the mitochondrial genome. Finally, the ER emerges as a hub that responds to internal and external stressors by regulating translation at the polysomes of the ER membranes. As such, we provide evidence for the role of metabolically active organelles as a hub for miRNA regulation to restore homeostasis.

## 6. Spatiotemporal Dynamics of miRNAs under Stress Conditions

Living cells have developed mechanisms to cope with stresses of many different types, including acute and chronic stress. Depending on tissue or organ, age, sex, medication load, and so on, cells are exposed to a diversity of stressors such as heat stress, hypoxia, oxidative stress, proteotoxicity, drug exposure, viral infection, and more. In reactions, cells that experience stress can activate processes such as apoptosis or autophagy. An emerging notion is that miRNAs are dynamically expressed and localized in response to environmental changes and to a variety of stress conditions [106]. How do miRNA numbers and locations change under dynamic conditions? What role does the ER play in the cellular stress response? How do miRNAs interact with other RNAs and RNA-binding proteins (RBPs) to concentrate the cellular stress response? How do viral miRNAs affect the host cell?

Our current knowledge on transcript localization is limited, relying mostly on in situ hybridization (ISH). This approach provides insights into the precise temporal and spatial control of gene expression [107,108], especially when used simultaneously with single-molecule imaging technologies to detect mRNAs and nascent proteins [109], as described in Section 3. In this section, we summarize the importance of mRNA localization as an overlooked aspect of gene expression. As a rule, regulation by miRNAs can occur only when the appropriate MSB is accessible, while the presence of other RNA-binding proteins will prevent such regulation [110]. Moreover, it was shown that in addition to the change in the competition of MBSs by RNA-binding proteins, the amounts of Ago2-bound miRNAs also mediate the translational capacity of the cellular stress response [111].

Figure 3 illustrates schematically how the distribution of multiple types of miRNAs among major organelles might be affected under stress, such as the use of a drug, oxidative stress, heat shock, etc. Monitoring the cellular localization of miRNAs governs their potential impact on cell physiology. For example, it was shown in HeLa cells that miR-484 translocates from the nucleoli to the nucleus and eventually to the cytoplasm following infection by the influenza A virus [41]. In addition, in human umbilical vein endothelial cells (HUVEC), miR-381-3p is redistributed between cytosol and mitochondria, whereas the overall level of this miRNA remains constant. Endothelial cell injury caused by reactive oxygen species is exacerbated by the miR-381-3p dynamic redistribution [112].

### 6.1. The ER as a Hub of miRNA-Mediated Stress Response

Cellular homeostasis is tightly linked to the cell’s ability to respond to a range of external stress conditions. As many pathologies are associated with the ER’s inability to dissipate oxidative and other stress types [113], we specifically focus on the ER stress response. We ask whether a change in the occurrence of specific miRNAs impacts ER functionality. Dysregulation of the ER leads to proteotoxicity and cell death, presumably as a failure of the unfolded protein response (UPR) [114]. The UPR in the ER manages proteotoxicity and misfolded proteins, resulting from acute or chronic ER stress that is caused by viral infection, redox stress, or bioactive damaging reagents [115]. The major branches of the UPR act in halting protein translation, degrading misfolded proteins, and activating pathways that produce molecular chaperones, each of which is tightly regulated by miRNAs [116]. Importantly, the importance of homeostasis maintenance by miRNAs is supported by the observation that many of the miRNAs that directly affect the UPR are differentially expressed during normal human aging [117,118]. The UPR is induced by BiP/GRP78 protein, the level of which is regulated by numerous miRNAs including miR-181, miR-30 family, miR-199a, miR-495, and miR-375. The other branches of the UPR are regulated by miR-30c, which targets *XBP1* [119]. Induction of UPR by external reagents (e.g., by tunicamycin, thapsigargin) leads to changes in the composition of tens of miRNAs [120]. Overall, the extensive involvement of miRNAs in UPR suggests that they act as agents for the stress responses, protecting cells from apoptosis and irreversible damage [121].

### 6.2. Stress-Related Membrane-Less Organelles

The cell houses several conserved ribonucleoprotein (RNP) granules, including Cajal bodies, nucleoli, stress granules (SGs), and processing bodies (P-bodies, PBs). Each of these granules has a distinct molecular composition and function, but they all comprise highly complex networks of protein–RNA interactions and low-complexity protein sequences. These are dynamic and reversible structures driven by the biophysical process of liquid–liquid phase separation (LLPS). SGs are formed under conditions such as heat stress, hypoxia, oxidative stress, and viral infection, which drive the formation of such liquid-like condensates [122,123]. The exact composition of SGs is not known. They were shown to be enriched in preinitiation complexes and include inactive ribosomal subunits and initiation factors, inhibitors of mRNA stability and translation, miRNAs, and Ago proteins. For example, miR-335 was shown to promote SG formation, and under hypoxia conditions during acute ischemic stroke, its levels were downregulated, and consequently, SG formation was attenuated [124].

In contrast to SGs, PBs are ribosome-free cytosolic compartments, containing factors involved in mRNA degradation and stability, such as decapping enzymes, miRNAs, and Ago proteins [125]. They were discovered during the investigation of the localization of proteins associated with the 5′-to-3′ mRNA decay pathway [126]. A recent analysis of PBs by following the dynamics of an individual miRNA molecule revealed that localization to PBs leads to unusual miRNA stability, supporting a hypothesis that PBs sequester unused miRNAs for surveillance, but not for mRNA decay [125]. The translation repression by miRISC is required for the delivery of mRNAs to PBs [127].

PBs and SGs are formed through LLPS, and mediated by intrinsically disordered proteins [128]. Under stress, Ago proteins localize to both SGs and PBs, but only for SGs does their localization require the mediation of miRNAs [129]. Ago was proposed to move between functional compartments such as the cytosol, nucleus, PBs, and SGs [130]. The formation of phase-separated droplets was shown to be formed by Ago2 and TNRC6B/GW182, a highly disordered core protein of the miRISC complex [131]. This higher-order complex increased its functionality and accelerated the deadenylation of mRNAs bound to the miRISC condensate.

In summary, PBs are considered the scaffolding of miRNA function, where the knockdown of GW182 disrupts PB formation. Based on innovative imaging studies, it became evident that mammalian PBs are primarily involved in the coordinated storage of mRNAs encoding regulatory functions rather than in RNA decay. The storage of mRNA is applied to thousands of transcripts, with about a third of all cellular mRNAs being enriched in PBs [132].

### 6.3. Viral Infection

In addition to the cells’ own set of miRNAs, they may also originate from viral infection. Following viral infection, cells undergo drastic changes due to an urgent need to reallocate resources. As miRNAs are involved in almost all cell processes, infection could be considered another type of stress in this regard. The genome of numerous mammalian DNA viruses contains small ncRNAs that can be further processed to produce functionally mature miRNAs that allow the infected cells to avoid the surveillance and action of the host immune system [133]. For example, the ncRNA transcribed from the *LAT* (latency-associated transcript) gene of herpes simplex virus type 1 (HSV-1) is processed to miRNA that suppresses TGF-β signaling. This results in the protection of infected neuronal cells from entering apoptosis [134]. There are 27 mature HSV-1 miRNAs that alter mostly viral transcripts but also act on cellular transcripts to promote viral replication, cell proliferation, and immune evasion [135]. In another example, the miRNA from simian-virus-40 (SV-40) leads to the downregulation of viral antigens, avoiding recognition of infected cells by cytotoxic T-cells, thus providing a strong advantage to the virus [136]. Furthermore, miR-155 and miR-223 expression are induced following infection with vesicular stomatitis virus (VSV), thus activating the type 1 interferon (IFN) response that mediates the inflammatory response. Alternatively, viral infection can inhibit the maturation of host miRNAs that are involved in the antiviral response by acting on RNA-binding proteins such as Dicer and Ago [137]. In addition to the canonical role of miRNA in RNA-mediated interference, in HIV-1, miRNAs interact with the viral Gag protein, preventing Gag from dimerizing at the plasma membrane. The end product is the inhibition of viral particle production [138].

In addition to viral miRNA within the cell, some viral miRNA affects intercellular communication through exosomal miRNAs. Both Kaposi sarcoma-associated herpesvirus (KSHV) and Epstein–Barr virus (EBV) encode miRNAs in their genomes. A study of lymphoma cell lines showed that 50% of the miRNAs in exosomes were derived from KSHV-infected cells. In fact, among KSHV-encoded miRNAs, many contained short sequences of CCCT and CCCG motifs that control the loading of miRNAs into exosomes [139]. Interestingly, the cases in which viral miRNA and host RNA interact to enhance miRNA degradation emphasize the continuous coevolution that fine-tunes the regulation of miRNA by external agents such as human DNA viruses [140].

### 6.4. Exosomes as Agents of Intercellular miRNA Regulation and Dynamics

In recent years, miRNAs have been detected in a variety of biological bodily fluids (e.g., serum, blood, and urine). In many cases, these miRNAs are located in exosomes, extracellular nuclease-resistant entities that are remarkably stable [141] and serve as a means of intracellular communication. Active miRNA sorting mechanisms were suggested for miRNA localization into extracellular vesicles [142]. The sources of the vesicles with miRNAs include apoptotic bodies, shed vesicles, and exosomes [108]. Under the scenario of exosomal communication, in the recipient cells, the expression of target genes can be attenuated. However, it remains debatable whether the numbers of miRNA found in exosomes are sufficient to impact the recipient cell in vivo [61]. Exosomes were found to mediate the transfer of miRNAs from T cells to antigen-presenting cells (APC) in the context of immune cell communication. The importance of the transferred miRNAs on the recipient cell was demonstrated [143].

Although exosomes play an important role in healthy cells and tissues, under stress, the composition and number of miRNAs transported through exosomes are changed, as demonstrated following viral infection. In addition, exosomes that carry miRNAs have the potential to regulate the oxidative stress response [108]. For example, in aging, oxidative stress causes the upregulation of miR-96, miR-182, and miR-183 in exosomes derived from bone marrow stromal cells (BMSCs), resulting in reduced osteogenic differentiation in BMSCs and reduced Hmox1 levels [144]. In major neurodegenerative diseases [145], the communication among neurons, microglia, astrocytes, progenitors, and endothelial cells is critical to executing the oxidative stress response [146]. Each cell type comprises a unique set of ncRNAs, including miRNAs, proteins, and lipids. Specifically, more than 100 miRNAs were detected in the exosomes from astrocytes. The abundant miR-873a-5p attenuated neuroinflammation by modulating the microglial NF-κB signaling pathway [147]. Within the brain, exosomes are constantly released and captured by neighboring cells. Such dynamics may alter the apparent miRNA composition of the recipient cells [141].

In summary, various stressors exert demands on the cell’s regulatory networks through the reorganization of the miRNA pool, among others through RNA–RNA and RNA–RBP interactions that occur in specialized cytosolic compartments. These dynamics are not limited to the cell’s own miRNAs, but also act upon, and are affected by, miRNAs from viruses as an external miRNA source. Finally, exosomal miRNAs may mediate intercellular communication and cell response.

## 7. Concluding Remarks

In this review, we aimed to provide an integrative view of spatial and temporal aspects of miRNA function under homeostasis and stress. These aspects are often overlooked in studies on specific miRNAs and their targets, whereas any cellular function requires strict regulation of the molecular interactions that determine these functions. Any future investigation on both the molecular level and at the tissue level requires an investigation of these spatial and temporal aspects. An increased understanding in this regard will serve the targeting of miRNA-mediated pathologies.

## Figures and Tables

**Figure 1 ijms-23-16152-f001:**
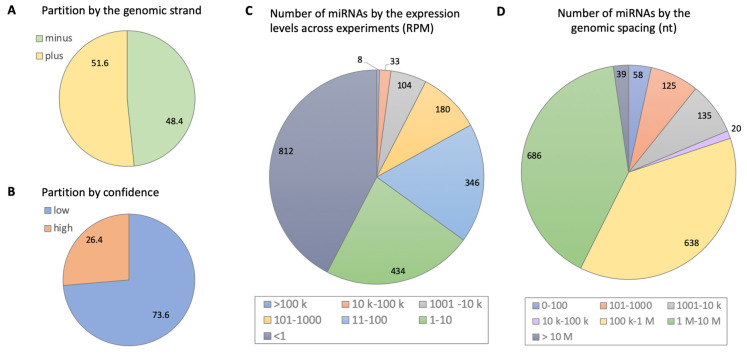
Summary statistics of human miRNAs according to data compiled in miRBase: (**A**) miRNA gene position by DNA strand. (**B**) Confidence level according to availability of experimental evidence. (**C**) Normalized levels of expression measured by reads per million (RPM) across the experiments compiled in miRbase. (**D**) The spacing of miRNA by genomic vicinity, measured by nucleotides (nt).

**Figure 2 ijms-23-16152-f002:**
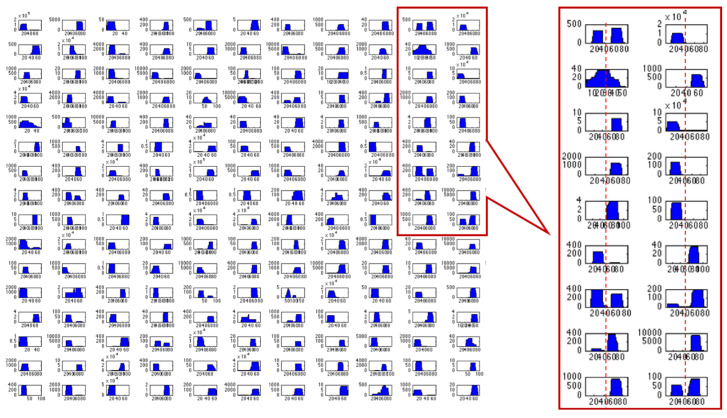
A collection of miRNAs expressed as reported in the miRBase database. Each panel shows the expression level of mature miRNAs by RPM (reads per million, y-axis). The active miRNA molecules are derived from the pre-miRNA, marked 0 to 80 nucleotides (x-axis). A zoom-in of a set of human miRNAs is marked by a red frame. Note: the broad range of RPM associated with individual miRNAs (y-axis).

**Figure 3 ijms-23-16152-f003:**
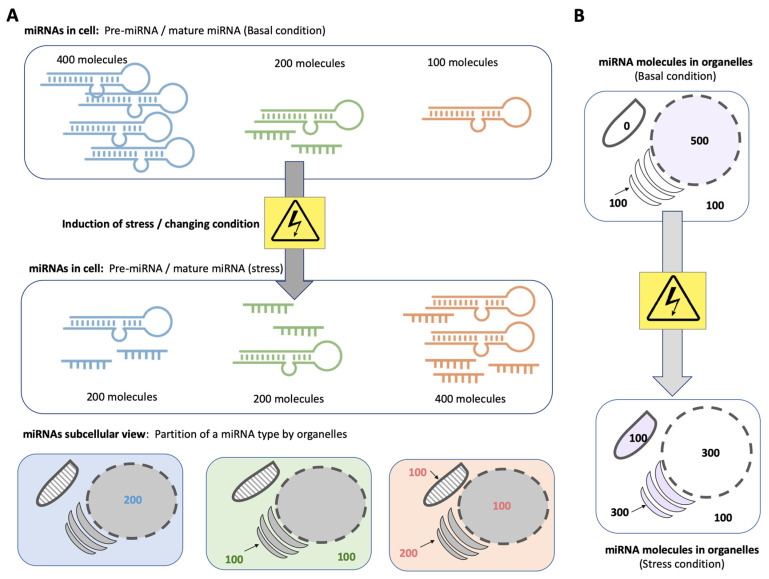
Illustration of a cell exposed to a stress condition: (**A**) The view of three types of miRNAs (denoted by their color), and the changes in the miRNA quantities and/or processing in response to stress. The subcellular localization of the miRNA molecules is illustrated. (**B**) A cellular view on miRNA localization. Organelles that are signified by maximal changes in miRNA number before and after external stress are colored.

## Data Availability

Not applicable.

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
