# Peer review of "Challenging Cellular Homeostasis: Spatial and Temporal Regulation of miRNAs"

_ijms, 2022, doi:10.3390/ijms232416152_

Round 1
Reviewer 1 Report
I think the authors did well with choosing to adapt this big topic into a readable article, however, for a review paper, there is less to learn and more to think about, as this article reminds me more of a philosophical and scientific textbook for undergrad style. The scope of the article is vast and due to this fact reader loses interest after the introduction/outline part. I would lose the outline section as it’s a distraction, maybe try to incorporate it into the introduction. Revise Figure descriptions in the text and under the figure, as both are quite long and some parts are duplicated – rather make a better explanation in the text and make a shorter one under the figure.
Some English style and formulation editing throughout the article is needed, as statements such as “we cite excellent sources” is unsuitable. Some sentences /information are repeated, do this only if this is of grave importance to the given section that it would lose its point.
Try doing a table for section 3 listing miRNAs and their roles/regulations/origins. Also, this section is quite long and lacks specifics, try with more examples.
Note, my comments are aimed to improve the readability and interest of future readers. Also, try to organize the article in a different way maybe even split the article into two separate parts? As the authors have the premise better in mind, I recommend for the team read the article and work on organization/reordering/readability. I see the value in the article, I just can’t imagine readers to finish reading this article with interest.
Also, I can’t leave out the excellent citing. For an almost 7 thousand word review of this topic, I would expect a few more references.
And separate abbreviations section is not necessary, explain abbreviations in the text.
Author Response
Reply: Thanks for highlighting several very important issues and providing suggestions. In response to some of the recommendations, we restructured the article and revised the manuscript substantially. Still, we believe that conceptual writing on this topic is helpful, especially for newcomers and experts in other, related fields (e.g., cell biology, miRNA, cell physiology etc).
The main changes we adopted in view of your comments are:
- We reorganized the chapters and the flow to make the ‘story’ more enjoyable (and easier) to read.
- We updated the title to reflect the content better
- We removed non-informative section titles (e.g. introduction, outline) and focused on a clear message for each section.
- We made sure that the Figures are discussed in the text, and we compressed the figure legends and removed duplications. We also removed the original Figure 4 which was suggested (by the other referee) as too general.
- According to the request of the other reviewer, we added many more specific examples of miRNAs beyond neuronal and cancer cellular systems..
- We removed the clinical aspect of exosome and miRNA use in diagnosis and therapy.
- We have limited the detailed discussion of miRNA dynamics in the pathological context. Instead, we illustrate our discussion of temporal and spatial regulation of miRNAs under stress and in cellular homeostasis.
- We have shortened the text by 25% (from 6670 to 5000) but kept the references rich enough.
We believe that following these changes, we were able to improve the delivery and make the reading more enjoyable, informative, and interesting.
Reviewer 2 Report
In my opinion general idea of the manuscript was correct. However despite the idea I am not sure if authors had a specific plan for this article. Introduction seems to be the best part of the whole article. Rest of the paper is very chaotic, very general, with very few good examples. Authors in majority of examples focused only on two aspects neurons/neural disease and embryonic development. And even embryonic development was barely described. Dozens of tissues and disease and authors describe only pancreatic cancer, brain tumor and brain. This is very poor quality review addressing probably less than 20 % of possible implication of spatial and temporal distribution of miRNAs. I not recommend this review for publication.
Please find my general comments in the order of appearance. Minor and less important issues that there were found in the text were neglected here.
Please, remember that 3'UTR is not only site for miRNA acton, there are many examples of 5" or even CDS binding sites. (introduction page 1 last sentence)
In outline, there is few sentence that are marely repetition of the information included in the introduction part (line 6-10 of outline). On the other hand other sentence are kind of introduction (like sentence in line 4-6). Altogether I don't understand what actually is outline here. As second paragraph is actually kind of results including Fig 2.
Last sentence of outline - I believe this is rather to o personal for a review.I would rahter say general stamtnet, such as "we cite a lot of articles, however we are aware not all of articles were cited".
The first sentence in the 3.1 paragraph do not provide any information and could be omitted/removed.
First sentence of the second paragraph on page 4 (very general sentences with no information can be removed)
Last several sentences on page 5 in paragraph 3.2 - many of sentences seems to be kind of space filling. They do not provide significant content and are rather general statments. It should be rather final summary used with summary of other paragraphs at the end of article – which actually do not exist. Other paragraphs seem to have no summary, and whole article is neither summarized.
Paragraph 3.2 - Since no citation or any reason to mention plants exist why to even put such sentence. This would be enough to say that miPEP exist in human and what they are. Story about plants is unnecessary here. Second thing why authors even mention about lncRNAs in mitochondria? as review is about miRNAs. And the last several sentences of paragraph 3.2 just before paragraph 3.3. - authors use general-statment mito-RNAs which is rathere a idea of this paragraph. So I would rather expect to see specific examples with names of mito-miRNAs and not see general stamtnets saying mito-miRNAs and e.g. s
paragraph 3.3 showed information about 3 compartments, nucleus, ER and mitochondria, while Fig 3 additionally meniones processing bodies. While no data about them could be found. On the other hand information about ER is mixed with the content of paragraph 3.4 which is dedicated to stress. And together with ER description there is a lot of attention paid to stress. If this is a case maybe it should be written, in more detail it would be discussed in next paragraph and should be moved into 3.4.
the pictograms indicating different stressors are rather strange. Does fire mean heat stress, an snow, cold stress? what would than antena signal meaning? And why Fig 3 is in part 3.3. while stressor are in separate paragraph 3.4
Chapter 4. Second paragraph - how this part about CAT-1 is related to stress? The whole description indicates localization asepct and no induction upon stress?
Part just before paragraph 4.1 - How full paragraph about embryo and oocytes is conected with the title of paragraph 4? And the part about ISH or single molecule imaging technologies in the last paragraph is repetition of the last paragaph on previous page 7.
in the whole paragraph about PB and SC there is no single example how or what miRNAs are commonly found in those strauctures. What stressors, how this affect function. This is just a general summary of miRNA loading to PB that could be found in dozen of reviews.
Figure 4 is very general and do not describe any particular mechanism or stressor, or example miRNA.
I am not sure why exosomes has been put into paragraph 4 into stressful conditions as they exist all the time in normal and physiological conditions as well.
In general exosome part somehow is dedicated in one page only to consideration in brain. This is very strange and very few good examples are shown there. With many general statments only.
And the last paragraph on page 9 is about BBB and neurodegenerative disease while suddenly the specific example of miRNAs on next page number 10, there is pancreatic cancer.
Last pragraph just before chapter 5 suddenly jumps into MSC with few sentences and no examples just general idea. The whole paragraph about exosomes is dedicated actually to one diseas Alzheimer. This part actuall has very poor link with the general idea of the review related to spatial and temporal miRNA expression. - how authors jump from specific function into the use of exosomes as diagnostic markers detected in blood.
On page 11 The whole paragraph occuping middle part of page 11 is almost in full dedicated to mRNA transport with absolutely no infromation of the role of miRNA in this process, no single example. And only on single sentence referes to miRNAs at the end with very blurry connection to the whole process of RNA transport and translation regulation. Actually part of next paragraph on the same page 11, indicate that miRNAs can be expressed in differet regions of neurons and be co-transported with RNA, which from on one page could be actually described in few sentences where the most critical and best fitting this concept would be last 4 sentences about miR-181a
5.2. why viral infection is not a stressor (while few paragraphs above it was mentioned as stressor) and now it is located in paragraph extreme translational demands...
I totally do not understand aim of the paragraph 5.2 the information provided here is such a mix. How this at all refere to the translational demands? There is information about exosomes that contain viral miRNA, there is a info about viral miRNa inhibiting infected cell gene expression. But the most important aspect I would say which would be rather host miRNA response to viral infection is compacted in 3 sentences juat mentioning processes- such as cell division, translation, morphology, inflamation. Nothing is described.
Author Response
Comments and Suggestions for Authors
In my opinion general idea of the manuscript was correct. However despite the idea I am not sure if authors had a specific plan for this article. Introduction seems to be the best part of the whole article. Rest of the paper is very chaotic, very general, with very few good examples. Authors in majority of examples focused only on two aspects neurons/neural disease and embryonic development. And even embryonic development was barely described. Dozens of tissues and disease and authors describe only pancreatic cancer, brain tumor and brain. This is very poor quality review addressing probably less than 20 % of possible implication of spatial and temporal distribution of miRNAs. I not recommend this review for publication.
Please find my general comments in the order of appearance. Minor and less important issues that there were found in the text were neglected here.
Reply: Thanks for highlighting critical issues. We clearly made effort to improve the flow, remove repetition, provide more concrete examples, and make the delivery more concise and focused.
We have not covered the entire cellular context in which miRNA localization and dynamics are fundamental (as in differentiation, embryogenesis, stem cell biology etc). Instead, we addressed the issue of cellular homeostasis in cells and the role of miRNA (controlled by time and space) to maintain and store it. We then provide a conceptual discussion on organellar functions that are regulated by miRNA (e.g., in nuclei, ER etc). We then discuss what happens under stress conditions (demonstrated PBs, SGs, exosomes). We illustrate viral infection as unique stress, where a shift in translation must occur. A special case of local translation in neurons is used as an example of the need for revisiting solutions in translation. We remove the section on neuronal and cancer pathology to provide a more focused summary.
More importantly, we made sure the story is well structured (and revised the order of the chapters accordingly). We added specific examples that were established for spatial/ temporal dimensions. We believe that following major rewriting, and restructuring of the manuscript, we were able to improve the delivery and make the reading enjoyable and informative.
Please, remember that 3'UTR is not only site for miRNA acton, there are many examples of 5" or even CDS binding sites. (introduction page 1 last sentence)
Reply: we have added a clarification in section 1 (titled “miRNA function at the right place and the right time")
In outline, there is few sentence that are marely repetition of the information included in the introduction part (line 6-10 of o utline). On the other hand other sentence are kind of introduction (like sentence in line 4-6). Altogether I don't understand what actually is outline here. As second paragraph is actually kind of results including Fig 2.
Reply: We have removed repetitions and reorganized the review. We also made sure to make the manuscript shorter (by 25%) and therefore much of the repetition was removed. Section 1 now serves as an introduction and presentation of the review’s scope.
Last sentence of outline - I believe this is rather too personal for a review.I would rahter say general stamtnet, such as "we cite a lot of articles, however we are aware not all of articles were cited".
Reply: We have removed it. Also, as it is more focused, there is not need to skip primary publications.
The first sentence in the 3.1 paragraph do not provide any information and could be omitted/removed.
First sentence of the second paragraph on page 4 (very general sentences with no information can be removed)
Reply: We have removed non-informative sentences.
Last several sentences on page 5 in paragraph 3.2 - many of sentences seems to be kind of space filling. They do not provide significant content and are rather general statments. It should be rather final summary used with summary of other paragraphs at the end of article – which actually do not exist. Other paragraphs seem to have no summary, and whole article is neither summarized.
Reply: We made sure to better summarize the sections. We added sentences that better connect the different sections and added a conclusion section (that was missing in the original version).
Paragraph 3.2 - Since no citation or any reason to mention plants exist why to even put such sentence. This would be enough to say that miPEP exist in human and what they are. Story about plants is unnecessary here. Second thing why authors even mention about lncRNAs in mitochondria? as review is about miRNAs. And the last several sentences of paragraph 3.2 just before paragraph 3.3. - authors use general-statment mito-RNAs which is rathere a idea of this paragraph. So I would rather expect to see specific examples with names of mito-miRNAs and not see general stamtnets saying mito-miRNAs and e.g. s
Reply: We have improved the part on miPEPs and removed referrals to plants as suggested. We added specific examples regarding mitomiRs to substantiate the messages.
paragraph 3.3 showed information about 3 compartments, nucleus, ER and mitochondria, while Fig 3 additionally meniones processing bodies. While no data about them could be found. On the other hand information about ER is mixed with the content of paragraph 3.4 which is dedicated to stress. And together with ER description there is a lot of attention paid to stress. If this is a case maybe it should be written, in more detail it would be discussed in next paragraph and should be moved into 3.4.
Reply: We are sorry that the chapters were not organized always logically, as you mention regarding the ER. Thanks for the reorganization suggestions. We adopted it as proposed. We have also revised the title of each section to better reflect its contents.
the pictograms indicating different stressors are rather strange. Does fire mean heat stress, an snow, cold stress? what would than antena signal meaning? And why Fig 3 is in part 3.3. while stressor are in separate paragraph 3.4
Reply: As a very schematic view of ‘stress’ by using simple icons. As it causes confusion, w have simplified the picture and removed the symbols.
Chapter 4. Second paragraph - how this part about CAT-1 is related to stress? The whole description indicates localization asepct and no induction upon stress?
Reply: Sorry for the unclear writing in the original version. CAT-1 mRNA is released from processing bodies under stress. We have clarified it in the manuscript.
Part just before paragraph 4.1 - How full paragraph about embryo and oocytes is conected with the title of paragraph 4? And the part about ISH or single molecule imaging technologies in the last paragraph is repetition of the last paragaph on previous page 7.
Reply: We remove all references to embryonic development to keep the paper focused. Evidently, that miRNA localization in the embryo provides crucial spatial information that dictates embryonic polarity and development program. We have also removed the repetition on ISH.
in the whole paragraph about PB and SC there is no single example how or what miRNAs are commonly found in those strauctures. What stressors, how this affect function. This is just a general summary of miRNA loading to PB that could be found in dozen of reviews.
Reply: Revised to add specific examples.
Figure 4 is very general and do not describe any particular mechanism or stressor, or example miRNA.
Reply: We deleted Figure 4 as it was too general.
I am not sure why exosomes has been put into paragraph 4 into stressful conditions as they exist all the time in normal and physiological conditions as well. In general exosome part somehow is dedicated in one page only to consideration in brain. This is very strange and very few good examples are shown there. With many general statments only.
And the last paragraph on page 9 is about BBB and neurodegenerative disease while suddenly the specific example of miRNAs on next page number 10, there is pancreatic cancer.
Reply: In accordance with your justified remark, we have clarified that despite a clear role for miRNA-mediated intercellular communication through exosomes in healthy cells and tissues, we have chosen to focus on exosomal localization and dynamics of miRNAs involved in the stress response. We have improved the focus of the paragraph and added specific examples of exosomal miRNAs in both neurons and other tissues under stress.
Last pragraph just before chapter 5 suddenly jumps into MSC with few sentences and no examples just general idea. The whole paragraph about exosomes is dedicated actually to one diseas Alzheimer. This part actuall has very poor link with the general idea of the review related to spatial and temporal miRNA expression. - how authors jump from specific function into the use of exosomes as diagnostic markers detected in blood.
On page 11 The whole paragraph occuping middle part of page 11 is almost in full dedicated to mRNA transport with absolutely no infromation of the role of miRNA in this process, no single example. And only on single sentence referes to miRNAs at the end with very blurry connection to the whole process of RNA transport and translation regulation. Actually part of next paragraph on the same page 11, indicate that miRNAs can be expressed in differet regions of neurons and be co-transported with RNA, which from on one page could be actually described in few sentences where the most critical and best fitting this concept would be last 4 sentences about miR-181a
Reply: Thank you for your remarks. As stated, the goal of our paper is to describe temporal and spatial aspects of miRNA function in homeostasis and the physiological stress and cell response. Although a vast body of research exists that investigates the function of miRNAs in disease, for the sake of focus and clarity, we have removed most disease-specific examples and other sidesteps which you mention in your comment (e.g. exosomes as diagnostic markers detected in blood).
5.2. why viral infection is not a stressor (while few paragraphs above it was mentioned as stressor) and now it is located in paragraph extreme translational demands...
I totally do not understand aim of the paragraph 5.2 the information provided here is such a mix. How this at all refere to the translational demands? There is information about exosomes that contain viral miRNA, there is a info about viral miRNa inhibiting infected cell gene expression. But the most important aspect I would say which would be rather host miRNA response to viral infection is compacted in 3 sentences juat mentioning processes- such as cell division, translation, morphology, inflamation. Nothing is described.
Reply: We have restructured the section on viral infection, with more focus on changes in cellular functions in response to viral infection. In addition, we have moved this part to the section on miRNA-mediated stress response as suggested. Overall, we incorporated substantial improvements to minimize ‘sidewalks. In addition to restructuring the sections, we provided more informative titles for each section.
Round 2
Reviewer 1 Report
I see a substantial change in the article, changes and shortening proved a good way to enhance clarity and message. I also like this new conception after rewriting.
However, there needs to be one more read-through – correcting English, grammar, e.g. line 24 „is govern by miRNA under“ – should be governed and there are more. Also figure 2 is duplicated, I don’t know if for the revisions tracking or if it's just duplicated. Also, all changes made to the article can be accepted and I would like to see the new form without deleted sentences (tracking is good for Word, where you can turn it off, but hideous in pdf) – this is not a failure on the authors’ side, but I can't read the article well in this form.
Author Response
Reply: Many thanks for your remarks. We have improved the structure and flow of the manuscript. As suggested, we briefly discuss the aspect of sponging by RNA-RNA interactions (e.g., by circRNAs and lncRNAs) and added missing data on the critical role in embryogenesis and maternal–zygotic transition. The technical issue (e.g., double appearance of Fig 2) was a leftover from the track change. We now, therefore, provide only a clean version to simplify the reading.

Reviewer 2 Report
Dear authors,
Reviesing most part of the manuscript within couple of days, can't give good results. I still do not think that authors make presentation of this idea to be clear and interesting. There is chaos and not fluent transition between different parts. Subject of separate paragprahs seems to be overlaping. I would summarize all of them as need for miRNA and mRNA to be in exact time and space. This is very often repeated. I believe that this is important aspect for consideration in studies, which authors try to point out as a problem however it is hard to conclude from the current manuscript, how to do this, when to do this etc. Authors in their workflow firstly indicate to consider spatial organization of miRNA and targets, their cellular localization (mitochondria or ER). Further they focus on exosomes as mediators of miRNAs that can affect distinct cells. And here actually this idea about spatial localization is somehow lost. We rather talk about cell to cell contact and effect of miRNA on other body compartments but not tissue or cellular compartments. In my opinion for the non familiar scientist it would be difficult to find clear answers in this review. after reading a title "challenging cellular homeostasis, spatial and temporal regulation of miRNAs".
I believe that authors should spend more time and re-think the concept of article.
There should be more direct questions and direct answers to them. I believe that in research where people find direct targets the difference in expression and changes in efficiency needs to be really great to be observed. Therefore total extracts from cells are giving good idea about targeting by miRNA. In most cases problem would be visible in finding targets when changes would be small, and therefore might be overlooked when not taking into accaunt spatial distribution. The big problem in miRNA research is also whether miRNA affect mRNA or protein level or both. What and when could be detected. In the manuscript I find several general assumptions, especially regarding the fact that we know about function of miRNA based on the 3'UTR interaction. I still believe that there might be other probem in miRNA research, which is that many interactions might be overseen or neglected due to lack of direct binding miRNA-target evidence as well as cumulative effect of regulation by miRNA as well other proteins, TF, or non-coding RNAs. And this issue was not considered here, or was barely touched in few places which made more confusion than real answers.
I do not recommend this article for publication.
There are many issues I would like to rise.
Line 34, I do not understand what authors mean by this. Knowledge of miRNA function is confined to molecular level? Is it? I do not think so. Actually majority of experiments first test the function, the effect of miRNA on the process, protein or RNA expression and one of the last aspects is related with finding the mechanism which is either miRNA-target interaction, or competition with non-coding RNAs or sponging.
Line 43 - this sentence sounds strange - how competition can be manisfested by lncRNA or circRNAs? I believe that I understand what authors try to say but this is not the way to do it. Morover no follow up on the idea of sponging miRNA by different factors has been done here.
Line 82 - is these not the same information as in the rest of sentences line 84-85. it sounds like miRNA has their own propmoters or are located in host genes, but miRNAs located with gost genes still have their own promoters? It is very blurry massage.
Line 150 -authors point out very important role of miRNAs in embryogenesis and do not follow this idea. in line 154 authors jump into p64 factor and show no name for any miRNA. In addition authors cite a review instaed of original work, and moreover most of examples in this review comes from mouse model. I believe that there would be plenty of exaples indicating role of miRNA in embryogenesis - including fertilization, role of miRNAs in sperm, oocyte, development of germ layers. From one site authors try to touch different aspects on the other hand jump from subject to subject such as epidermis or cholesterol.
Part between line 161 and 169. Authors introduce lncRNA, circRNA and give no examples no interaction with miRNAs and jump into next paragraph which is about motor proteins and miRNA redistribution.
line 186 why it is metioned about pre-miR-338 to have effect on cytochrome c? If mature forms are active regulators of cell fate? I can see the point of information in line 186-200 but the massage is somehow chaotic. Authors describe, pri-miRNAs, pre-miRNAs and where are mature miRNAs in this story. There is not much logic here. It should be clearly stated increased and localized levels of pri and pre miRNAs, coincidence with high local levels of mature miRNAs which can exert they function. If this is at al a case? Unless authors describe process in which pre-miRNA regulates COXIV expression??
Titles are still not fitting the main context of each paragraph. Paragraph 4 is entitled co-localization of miRNAs and targets, while most of the paragraph 3 was actually saying that miRNA and mRNA must be in same place such as line written in previous paragraph in line 195, which also makes paragraph 3 title not very accurate.
326-344 lines are actually parts that could fit well to this paragraph. However still authors indicate in introduction that they will focus on plethora of miRNAs found in human, while plenty of examples are shown in mice or as for miR-9 shown example for zebrafish. Taking into account high similarity of many miRNAs between species it could be extrapolated to the human however this link or at least some idea is missing. From the original paper e.g. for miR-9 line 340, we can conclude that this miRNA in general is important for brain however specific localization into nucleus results in stemness of NPSC, which actually is changed depending on the proportion of miR-9 in cytoplams and nucleus. And it is hard to follow this based on a verry short description used by authors.
line 401-403 uthors are citing one of the interesting papers showing role of so called miPEP-133 with no details about that in their text. Morover many of statments used by authors are very simplified making confusion. Authors write "some miRNAs are translated to peptides" - which is not a case. If we go into detail we can read in origin papers more in detail that some miRNA encoding genes can have open reading frame fragments, around the original miRNA sequence. And such mir-PEP 133 originating from miR34 locus contains 402 bp ORF, giving rise to 133 aminoacid miR-PEP, which not at all is miRNA translated to protein (citation 86). And dispite that authors indeed removed word plants form text still the rest of examples of miR-PEP actions cites plant research.
Line 500. Authors suggest that miR-204 is located within ER? Citated papers, and short literature review indicates actually that indeed Caveolin and Sirtulin might be responsible for ER stress mediated by miR-204. However most authors indicate that Caveolin 1 (Cav1) is " expressed in the plasma membrane caveolae of the endothelial cells and has an important role in the regulation of vascular function". There are also evidence that Cav-1 is a cholesterol-binding protein that can transport cholesterol from the endoplasmic reticulum to the plasma membrane. However it is hard to assume weather ER induced stress is due to miR-204 action within ER or blocking levels of Cav1 around cell membrane or in cytoplasm. No direct eveidence for its role. Unless authors can provide examples of such explanataions that this action is indeed in ER, and not only effect is related with ER stress but action happens elsewhere.
I find somehow interesting the figure where authors try to say that not only total miRNA level but specific/local expression or localization might be changed, showing redistribution among compartments (Fig3). However it still not very well illustrated by citations and research work. Or maybe nobody has provided spatial and temporal redistribution of miRNAs among diverse subcellular localizations?
Lines 534-540, why this specific example is actually used here in the very general introduction to the paragraph 5. And despite different titles starting from paragraph 3 via 4 and 5, the idea is the same and is repeated that miRNA and mRNA need to be in the same place. And example of CAT1 is actually showing competition between different molecules targets.
Figure with the nucleus and cytoplasm and miRNA processing. In nucleus there is pre-miRNA as well as in cytoplasm. Should not be one pri-miRNA?
paragraph 5.3 majority of paragraph is very general. I would expect more examples of exosome derived miRNA action in other cells, which is in the end of it. Why authors focused only on oxidative stress as the only example. What cells and how they communicate. There are plenty of great evidence that miRNA or other RNAs sent via exosomes can affect other cells - also in tumors or immune cells. If already authors say about serum or blood, are there more evidence for distant regulation of targets (good example provided by authors is miR-137 in PD, however further implication of OXR1 in oxidative stress is rather unnecessary). And the same refers to Line 911/912 about SIRT1, which is not needed.
Author Response
Reply: The reviewer has raised substantial criticism regarding the clarity of the messages. Furthermore, the reviewer claims that critical challenges in the miRNA field remained unanswered, i.e., degradation of mRNA vs translation inhibition; the binding to 3’-UTR; the unseen effect of miRNA via indirect effects (by TFs, ncRNAs etc). We fully agree that the manuscript ignores many ‘big issues’ that were covered by 10s of excellent review articles. Instead, herein we intended to discuss much-overlooked aspects of time and space in miRNA regulation and the impact of cellular homeostasis.
Based on the critical comments, we have restructured the manuscript, removed repetitions, and added missing information. We have restructured and improved the flow and the case study of neurons and exosomes were removed to separate “Box” sections. We provide evidence that the time and space of miRNAs aspect of miRNA cell regulation dominates especially under stressful conditions. We have also added clear ‘questions’ to each section to better explain the content of each section and rephrased the title. We also deemed it important to provide a practical context for the reader, and therefore we have added a short section on experimental approaches to study spatiotemporal aspects of miRNAs (Section 3, revised). We have improved delivery of the messages by addressing the excellent and detailed comments that the reviewer provided us with. We hope that the reviewer appreciates the contribution of these aspects to the miRNA field.
There are many issues I would like to rise
Line 34, I do not understand what authors mean by this. Knowledge of miRNA function is confined to molecular level? Is it? I do not think so. Actually majority of experiments first test the function, the effect of miRNA on the process, protein or RNA expression and one of the last aspects is related with finding the mechanism which is either miRNA-target interaction, or competition with non-coding RNAs or sponging.
Reply: Thank you. We have removed this (confusing) statement and provided accurate description to experimental approach and the notion of sponging and competition.
Line 43 - this sentence sounds strange – how competition can be manisfested by lncRNA or circRNAs? I believe that I understand what authors try to say but this is not the way to do it. Morover no follow up on the idea of sponging miRNA by different factors has been done here.
Reply: We have rephrased Section 1. We included specific examples for the concept of sponging in Section 4.
Line 82 - is these not the same information as in the rest of sentences line 84-85. it sounds like miRNA has their own propmoters or are located in host genes, but miRNAs located with gost genes still have their own promoters? It is very blurry massage.
Reply: We have rephrased and clarified this section.
Line 150 -authors point out very important role of miRNAs in embryogenesis and do not follow this idea. in line 154 authors jump into p64 factor and show no name for any miRNA. In addition authors cite a review instaed of original work, and moreover most of examples in this review comes from mouse model. I believe that there would be plenty of exaples indicating role of miRNA in embryogenesis - including fertilization, role of miRNAs in sperm, oocyte, development of germ layers. From one site authors try to touch different aspects on the other hand jump from subject to subject such as epidermis or cholesterol.
Reply: Thank you for your comment. We have revised the section that analyze the importance of miRNA in embryogenesis (emphasizing the use of model organisms). Concomitantly, we have added specific examples of miRNAs that regulate processes to retain homeostasis, such as stemness, proliferative potential, cell identity, and more. We kept examples of metabolic homeostasis. It is apparent that healthy and diseased cell states often reflect a shift in miRNA metabolic homeostasis.
Part between line 161 and 169. Authors introduce lncRNA, circRNA and give no examples no interaction with miRNAs and jump into next paragraph which is about motor proteins and miRNA redistribution.
Reply: We have added a short description and examples of the concept of miRNA sponging in Section 4 (paragraph 3), explaining the role of competing endogenous RNA (ceRNA) by circRNAs and lncRNAs. We have removed the description for neurons as a polarized cell to Box 1, and for clarity we removed the miRNA connection to motor proteins.
line 186 why it is metioned about pre-miR-338 to have effect on cytochrome c? If mature forms are active regulators of cell fate? I can see the point of information in line 186-200 but the massage is somehow chaotic. Authors describe, pri-miRNAs, pre-miRNAs and where are mature miRNAs in this story. There is not much logic here. It should be clearly stated increased and localized levels of pri and pre miRNAs, coincidence with high local levels of mature miRNAs which can exert they function. If this is at al a case? Unless authors describe process in which pre-miRNA regulates COXIV expression??
Reply: Of course, there is no basis to suggest that pre-miRNA (or pri-miRNA) act like mature miRNA in attenuating gene expression. We have clarified this section accordingly. The locally matured miRNAs are the regulators of mRNA and we have provided several examples to demonstrate the capacity of local reservoir to provide function in the relevant part of neuronal cells. We added a simple scheme and moved specific examples to a Box 1 for highlighting the uniqueness of neurons in miRNA spatial regulation.
Titles are still not fitting the main context of each paragraph. Paragraph 4 is entitled co-localization of miRNAs and targets, while most of the paragraph 3 was actually saying that miRNA and mRNA must be in same place such as line written in previous paragraph in line 195, which also makes paragraph 3 title not very accurate.
Reply: Thank you for your comment. We have changed the section titles to better reflect the section’s contents along with the restructure of the manuscript
326-344 lines are actually parts that could fit well to this paragraph. However still authors indicate in introduction that they will focus on plethora of miRNAs found in human, while plenty of examples are shown in mice or as for miR-9 shown example for zebrafish. Taking into account high similarity of many miRNAs between species it could be extrapolated to the human however this link or at least some idea is missing. From the original paper e.g. for miR-9 line 340, we can conclude that this miRNA in general is important for brain however specific localization into nucleus results in stemness of NPSC, which actually is changed depending on the proportion of miR-9 in cytoplams and nucleus. And it is hard to follow this based on a verry short description used by authors.
Reply: We have updated the examples to include either cases of human miRNAs or cases in which results from animal models can be extrapolated to humans. We made effort to avoid confusion and rewrote the section on miRNA role in the nucleus.
line 401-403 uthors are citing one of the interesting papers showing role of so called miPEP-133 with no details about that in their text. Morover many of statments used by authors are very simplified making confusion. Authors write "some miRNAs are translated to peptides" - which is not a case. If we go into detail we can read in origin papers more in detail that some miRNA encoding genes can have open reading frame fragments, around the original miRNA sequence. And such mir-PEP 133 originating from miR34 locus contains 402 bp ORF, giving rise to 133 aminoacid miR-PEP, which not at all is miRNA translated to protein (citation 86). And dispite that authors indeed removed word plants form text still the rest of examples of miR-PEP actions cites plant research.
Reply: Thank you for your comment, we have improved wording and present human miPEPs to better support our discussion of this topic.
Line 500. Authors suggest that miR-204 is located within ER? Citated papers, and short literature review indicates actually that indeed Caveolin and Sirtulin might be responsible for ER stress mediated by miR-204. However most authors indicate that Caveolin 1 (Cav1) is " expressed in the plasma membrane caveolae of the endothelial cells and has an important role in the regulation of vascular function". There are also evidence that Cav-1 is a cholesterol-binding protein that can transport cholesterol from the endoplasmic reticulum to the plasma membrane. However it is hard to assume weather ER induced stress is due to miR-204 action within ER or blocking levels of Cav1 around cell membrane or in cytoplasm. No direct eveidence for its role. Unless authors can provide examples of such explanataions that this action is indeed in ER, and not only effect is related with ER stress but action happens elsewhere.
Reply: You are correct that miR-204 (or other miRNAs) are not is located to the ER. However, polysomes on the rough ER membrane act as hubs of RNA processing and regulation, which is reflected in the fact that in pull-down and fractionation experiments, RISC factors, including miRNAs and Ago proteins, are co-localized to the rough ER membrane. We have adapted the section to reflect this more accurately.
I find somehow interesting the figure where authors try to say that not only total miRNA level but specific/local expression or localization might be changed, showing redistribution among compartments (Fig3). However it still not very well illustrated by citations and research work. Or maybe nobody has provided spatial and temporal redistribution of miRNAs among diverse subcellular localizations?
Reply: Per your suggestion, we have added some examples of miRNA redistribution dynamics which is illustrated in Fig. 3. We revised the figure to also indicate that the amounts of mature / pre-miRNA is also relevant and define by cell type and the nature and duration of the stress condition. In addition, we have moved the figure to Section 6 which describes the dynamics of miRNA localization under stress.
Lines 534-540, why this specific example is actually used here in the very general introduction to the paragraph 5. And despite different titles starting from paragraph 3 via 4 and 5, the idea is the same and is repeated that miRNA and mRNA need to be in the same place. And example of CAT1 is actually showing competition between different molecules targets.
Reply: We have removed this example, as it is indeed not fitting in this introductory part of Section 6. Regarding section titles, we have updated them to better reflect their contents.
Figure with the nucleus and cytoplasm and miRNA processing. In nucleus there is pre-miRNA as well as in cytoplasm. Should not be one pri-miRNA?
Reply: This figure is illustrative only and the scheme refers to mature and non-mature version of the transcribed miRNAs. We have thus simplified the figure by only showing the numbers rather than the types of the miRNA-derived sequences.
paragraph 5.3 majority of paragraph is very general. I would expect more examples of exosome derived miRNA action in other cells, which is in the end of it. Why authors focused only on oxidative stress as the only example. What cells and how they communicate. There are plenty of great evidence that miRNA or other RNAs sent via exosomes can affect other cells - also in tumors or immune cells. If already authors say about serum or blood, are there more evidence for distant regulation of targets (good example provided by authors is miR-137 in PD, however further implication of OXR1 in oxidative stress is rather unnecessary). And the same refers to Line 911/912 about SIRT1, which is not needed.
Reply: The presence of miRNAs in exosomes (and in serum) is an active research area of translational medicine. We have moved the section on exosomes to a separate ‘Box’ to accentuate that miRNA relocation through exosomes is a special case of miRNA spatiotemporal dynamics. We have changed the examples in this section (as suggested) to represent a broader set of stress types.
Round 3
Reviewer 1 Report
I think the article has changed sufficiently for publication.
Author Response
thanks a lot for the support and the professional help. The last version will be uploaded in 2 days.
Reviewer 2 Report
I really appreciate effort that authors put into upgrade of the manuscript version currently submitted to the review. It was significantely improved, and much better and clearly described. One of the biggest problem that occures since the beginning is the editing - current version is lacking lines numeration and the worst is the fact that all new lines are mixed with crossed-out lines. The best (worst example is page 7, where in the whole page of paragraph 4 of crossed-out text there is one tiny sentence on the bottom of the page). This could be treated rather as authors draft and not version submitted to the review. Despite this I undertook the effort to read the manuscript, and I believe that the soudness and clarity is much better. However a lot of attention need to be paid and carefull editing of sentences need to be done, as many of words are crossed-out, and some are added, which could result in some errors.
I really like the idea to put some questions at the beginning of each paragraph, which puts some emphasize on what would be the subject in further lines. On the other hand I don't understand the idea to create Box1 and 2. This is actually to big part of manuscript. I do not feel as this is a good place. There is no particular reason or explenation why this two special cases are separately put into boxes. Please try to fit those examples into other pragraphs or create other new paragraphs. Or make the specific explanation for showcasing those examples as separate entities??
Issues that I will rise here are in the order as they appear in the uploaded version of the pdf.
- End of the page 3 and begining of page 4, which discribes diversity of miRNA. Actually in 3 sentences word diversity is repeated constantly. Few sentences futher there is again diversity repeated several times. Please try to not overuse this word or just use other.
Figure 2 on page 5 is very hard to read, very small, very compacted. Not sure what exactly they are showing. Is a one small box expression for sepcific miRNA, or tissue or cell?
There are some repetitions, therefore please double check for such "noise" - eg. on page 8, in two last paragraphs there is similar sentence "The importance of specific miRNAs was demonstrated to regulate cellular identity, for example....." and further in last bottom paragraph first sentence "The miRNA profile is established to maintain cellular identity and cell state [8]"
Page 10 repetition "the binding of miRNA binding sequences..."
Page 11. Still authors did not change this shortcut "Some miRNAs are translated to peptides (miRNA-encoded peptides, miPEPs) [98]". As I previously indicated original article do not say that miRNA is translated to protein but miRNA encoding gene contains open reading frames to code for proteins. Those are two different things. What authors write can be read as 21 nt miRNA give rise to 7 aminoacid protein? Please modify.
Page 12 I understand that the described paragraph related with stress tightly connects with ER function however it does not smoothly relates to the 5.3 ER paragraph. I would consider to create additional paragraph as author said that they focus on stress related changes in miRNA moe it probably to the paragraph 6 which is related to stress. Just leave the summary in part 5 "In summary, miRNAs located in close proximity of the nucleus, mitochondrion, and ER allow to fulfill specific functions that are absent in the cytoplasm. The nucleus is not only the site of miRNA transcript...."
Page 13: "miRNAs remain active and dynamically respond to environmental changes", this seems likek a shortcut. miRNAs repond? rather levels, transcription of miRNAs is addapted, miRNAs, genes are not leaving to be active and respond.
Figure 3. authors say "Figure 3 illustrates schematically how the distribution of multiple types of miRNAs among major organelles is affected under stress", as it was true and happened. This might happen or is possible to happen. Please rephrase this part.
"6.1. Membrane-less organelles". This is still unclear if this should be a part of organelles such as ER, mitochodnria or be kept here in paragraph about stress. I would re-name this as "stress related organelles" and could be probably solving the problem?
In general I believe this is much better version of the previous manuscript but still authors need to work on several issues rised here. I think that this article may have chance to be publihsed, however for the current review version I would say it needs major correction.
Author Response
I really appreciate effort that authors put into upgrade of the manuscript version currently submitted to the review. It was significantely improved, and much better and clearly described. One of the biggest problem that occures since the beginning is the editing - current version is lacking lines numeration and the worst is the fact that all new lines are mixed with crossed-out lines. The best (worst example is page 7, where in the whole page of paragraph 4 of crossed-out text there is one tiny sentence on the bottom of the page). This could be treated rather as authors draft and not version submitted to the review. Despite this I undertook the effort to read the manuscript, and I believe that the soudness and clarity is much better. However a lot of attention need to be paid and carefull editing of sentences need to be done, as many of words are crossed-out, and some are added, which could result in some errors.
Reply: We are sorry for the ‘technical’ difficulty. We have provided clean version to make it easy to read. However, by the editorial formatting the text to a ‘journal style’, we also noted that the text is quite hard to read and the line order becomes messy. Thanks again for putting the effort throughout the process.
I really like the idea to put some questions at the beginning of each paragraph, which puts some emphasize on what would be the subject in further lines. On the other hand I don't understand the idea to create Box1 and 2. This is actually to big part of manuscript. I do not feel as this is a good place. There is no particular reason or explenation why this two special cases are separately put into boxes. Please try to fit those examples into other pragraphs or create other new paragraphs. Or make the specific explanation for showcasing those examples as separate entities??
Reply: Actually, the two Boxes were added to allow a smother reading. With such Boxes, readers can either go through the Box, or not, without losing the flow of the manuscript. We can easily fit tease two elements in the current section (it will help in Reference numbering). In this current version this is what we did.
Issues that I will rise here are in the order as they appear in the uploaded version of the pdf.
- End of the page 3 and begining of page 4, which discribes diversity of miRNA. Actually in 3 sentences word diversity is repeated constantly. Few sentences futher there is again diversity repeated several times. Please try to not overuse this word or just use other.
Reply: Indeed, we have improved writing and avoid overuse of the word diversity.
Figure 2 on page 5 is very hard to read, very small, very compacted. Not sure what exactly they are showing. Is a one small box expression for sepcific miRNA, or tissue or cell?
Reply: We increased the fonts and added explanations to improve clarity.
There are some repetitions, therefore please double check for such "noise" - eg. on page 8, in two last paragraphs there is similar sentence "The importance of specific miRNAs was demonstrated to regulate cellular identity, for example....." and further in last bottom paragraph first sentence "The miRNA profile is established to maintain cellular identity and cell state [8]"
Reply: We sharpened the messages and removed repetition.
Page 10 repetition "the binding of miRNA binding sequences..."
Reply: Rephrased
Page 11. Still authors did not change this shortcut "Some miRNAs are translated to peptides (miRNA-encoded peptides, miPEPs) [98]". As I previously indicated original article do not say that miRNA is translated to protein but miRNA encoding gene contains open reading frames to code for proteins. Those are two different things. What authors write can be read as 21 nt miRNA give rise to 7 aminoacid protein? Please modify.
Reply: Thank you for this remark. We have improved our phrasing (line 309).
Page 12 I understand that the described paragraph related with stress tightly connects with ER function however it does not smoothly relates to the 5.3 ER paragraph. I would consider to create additional paragraph as author said that they focus on stress related changes in miRNA moe it probably to the paragraph 6 which is related to stress. Just leave the summary in part 5 "In summary, miRNAs located in close proximity of the nucleus, mitochondrion, and ER allow to fulfill specific functions that are absent in the cytoplasm. The nucleus is not only the site of miRNA transcript...."
Reply: Per your remark, we have moved this part on the ER stress response to chapter 6 to better align the messages conveyed in each chapter.
Page 13: "miRNAs remain active and dynamically respond to environmental changes", this seems likek a shortcut. miRNAs repond? rather levels, transcription of miRNAs is addapted, miRNAs, genes are not leaving to be active and respond.
Reply: Rephrased.
Figure 3. authors say "Figure 3 illustrates schematically how the distribution of multiple types of miRNAs among major organelles is affected under stress", as it was true and happened. This might happen or is possible to happen. Please rephrase this part.
Reply: Rephrased.
"6.1. Membrane-less organelles". This is still unclear if this should be a part of organelles such as ER, mitochodnria or be kept here in paragraph about stress. I would re-name this as "stress related organelles" and could be probably solving the problem?
Reply: We have renamed this section “Stress-related membrane-less organelles” to reflect the fact that we discuss specifically membrane-less foci in the cytoplasm, as opposed to membrane-bound, stable organelles. As mentioned in reply to your comment above, we have moved the section on the role of miRNAs in the ER-stress response to this chapter.
In general I believe this is much better version of the previous manuscript but still authors need to work on several issues rised here. I think that this article may have chance to be publihsed, however for the current review version I would say it needs major correction.
Reply: Thanks a lot for the effort that improved the manuscript substantially in content and structure. We appreciate the effort and suggestions throughout the process.
Round 4
Reviewer 2 Report
I belive current version of manuscript has been substantally updated. However, I would reccomend english correction as it could improve the quality of language to better deliver the message.